# FedSSP: Federated Graph Learning with Spectral Knowledge and Personalized Preference

**Zihan Tan**[1*]     **Guancheng Wan**[1*]     **Wenke Huang**[1*]     **Mang Ye**[1,2†]

[1] National Engineering Research Center for Multimedia Software, Institute of Artificial Intelligence,
Hubei Key Laboratory of Multimedia and Network Communication Engineering,
School of Computer Science, Wuhan University, Wuhan, China.
[2] Taikang Center for Life and Medical Sciences, Wuhan University, Wuhan, China
`{zihantan,guanchengwan,wenkehuang,yemang}@whu.edu.cn`

## Abstract

Personalized Federated Graph Learning (pFGL) facilitates the decentralized training of Graph Neural Networks (GNNs) without compromising privacy while accommodating personalized requirements for non-IID participants. In cross-domain scenarios, structural heterogeneity poses significant challenges for pFGL. Nevertheless, previous pFGL methods incorrectly share non-generic knowledge globally and fail to tailor personalized solutions locally under domain structural shift. We innovatively reveal that the spectral nature of graphs can well reflect inherent domain structural shifts. Correspondingly, our method overcomes it by sharing generic spectral knowledge. Moreover, we indicate the biased message-passing schemes for graph structures and propose the personalized preference module. Combining both strategies, we propose our pFGL framework **FedSSP** which **S**hares generic **S**pectral knowledge while satisfying graph **P**references. Furthermore, We perform extensive experiments on cross-dataset and cross-domain settings to demonstrate the superiority of our framework. The code is available at `https://github.com/OakleyTan/FedSSP`.

## 1 Introduction

Graph Neural Networks (GNNs) [56, 63, 48, 42] have demonstrated their superiority in modeling graph data which frequently emerges in a variety of scenarios [73, 71], as exemplified by graph clustering [45, 46, 44], graph contrastive learning [64, 80], anatomy detection [75, 88], knowledge graph [79, 78, 38], structural inference [66, 68, 65, 67] and so on. However, large amounts of graph data are generated by edge devices in reality, which brings in privacy concerns and the challenges of data silos [89, 24, 28]. To address these difficulties, federated learning (FL) has recently been applied to graph learning [18, 20, 25]. It allows models on various clients to collaborate without sharing local data [26, 15, 23, 27, 83] and makes federated graph learning (FGL) a promising direction. Nonetheless, the non-IID problem remains a major challenge in FGL, as graph data from different clients usually vary significantly. In such scenarios, a single global model struggles to adapt well to the local data of each client with inconsistent data distributions [60, 58]. To tackle these challenges, personalized federated graph learning (pFGL) has emerged, offering customized GNNs for each client to achieve satisfying local performance [1, 61].

However, pFGL still encounters substantial challenges from structural heterogeneity [29], especially in domain shift tasks, for instance, between social networks [94, 95] and molecular structures [59, 55]. There are two significant drawbacks to previous algorithms as Fig. 1 demonstrates. For global

---

*Equal contribution. † Corresponding author.

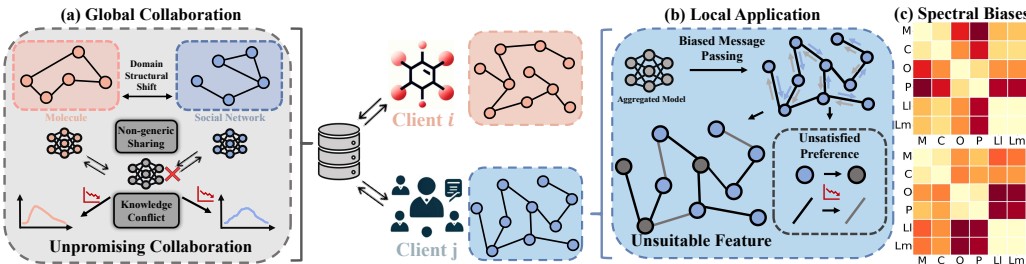

Figure 1: **Problem illustration.** We illustrate the challenges of the cross-domain scenario. (a) Considering the domain structural shifts, clients struggle with **knowledge conflict** caused by non-generic sharing which arises from the shifts, thus leading to unpromising global collaboration. (b) The aggregated message-passing scheme suffers from **inconsistent preferences** that remain **unsatisfied** of specific datasets in this scenario. Consequently, it leads to unsuitable features of graphs in local applications. (c) The heat map of Jensen-Shannon divergence of algebraic connectivity [17] and eigenvalues distributions among six datasets from three different domains. Spectral characteristics exhibit significant **biases across** domains but are more **similar** within a **same** domain.

collaboration, the considerable domain structural shifts inevitably lead to non-generic knowledge, thus resulting in knowledge conflict. Both current methods suffer from conflict and are trapped in unpromising collaboration. Specifically, [61] share non-generic structural encoding and struggle with structural knowledge conflict, while strategy [77] mitigating conflicts by limiting the potential for collaboration. The key to addressing knowledge conflict is pursuing a way to share generic knowledge that benefits all clients. Based on this observation, we raise the question: 1) *how to address the* **knowledge conflict** *under domain structural shift by extracting and sharing generic knowledge?*

For local applications, each client owns its specific dataset with distinct structural characteristics in this cross-dataset scenario. Due to the GNN message-passing nature, distinct graph structures stored in different clients prefer different message-passing schemes. Consequently, the scheme provided by aggregated GNN exhibits biases from the optimum when applied locally, thus leading to unsuitable features. Both current methods neglect the preferences of various clients for specific graph structures. This deficiency leads us to consider: *2) how to design personalized plans to deal with* **inconsistent preferences** *of specific graph datasets from various clients?*

To address problem 1), given that structural shifts make it hard to directly achieve generic sharing at the structural level, we propose to explore the structure shifts from another spectral perspective since previous works have demonstrated the strong correlation between graph structure and spectra [2, 81, 43, 32]. The major advantage of spectra is the detailed propagation and processing of graph signals on the graph structure, thus facilitating the discovery of generic knowledge in several certain processes unaffected by structural shifts. To validate our assumption, we first conduct experiments and explicitly reveal the domain spectral biases that directly reflect domain structural shifts on spectra as Fig. 1 demonstrates. To tackle these spectral biases directly to overcome structural shifts, we design **Generic Spectral Knowledge Sharing** (GSKS) to share generic spectral knowledge extracted from spectral encoders. It enables clients to benefit from others through collaboration without knowledge conflict. Conversely, other components containing non-generic knowledge are retained locally. Correspondingly, clients can customize powerful graph convolutions for their local graph characteristics while benefiting from generic knowledge without conflict. Through this strategy, we promote the sharing of generic spectral knowledge and the personalizing of non-generic knowledge, thus achieving effective collaboration against knowledge conflict.

Moreover, we attempt to achieve target 2) and design suitable personalized plans for each client graph structure locally. Specifically, we explore the message-passing nature of GNN [5, 16, 62]. From the spectral perspective, spectral encoders strongly affect message transmission. Therefore, when aggregated spectral encoders are applied to distinct graph structures locally, they tend to deviate from the optimal message-passing scheme for the client [49]. Consequently, GNNs extract inappropriate frequency messages which lead to unsuitable features. To meet the inconsistent graph preferences, we innovatively configure a learnable preference for each client and propose **Personalized Graph Preference Adjustment** (PGPA). These personalized preference modules apply adjustments to the feature extracted with the participation of global spectral encoders. It allows the feature to suit the specific graph structures of each client. Moreover, to address the issue of over-reliance when applying the preference module independently, a regularization term is introduced. Combining both strategies

for effective global collaboration and personalized local application, we propose our pFGL framework FedSSP. In conclusion, our key contributions are:

- We are the first to reveal domain structural shifts through spectral biases, as well as consider the inconsistent preferences of distinct datasets from various clients.

- We propose FedSSP, which innovatively overcomes knowledge conflicts from a spectral perspective and implements personalized graph preference adjustments for each client.

- We conduct experiments in various cross-dataset and cross-domain settings, proving that our approach outperforms several current state-of-the-art methods and achieves optimal results.

## 2   Related Work

### 2.1   Spectral GNNs

Spectral GNNs [4, 12] are based on spectral graph filters set in the spectral domain, providing powerful models for graph neural networks [76, 87, 86, 47, 72, 10, 70, 8, 69, 7]. Spectral GNNs can generally be categorized into two types: those with fixed filters and those with learnable filters. Fixed filter spectral GNNs, such as APPNP [19], utilize personalized PageRank (PPR) [53] to construct their filtering functions. GNN-LF/HF [96] designs filter weights from the perspective of graph optimization functions. Learnable filter spectral GNNs include subclass that approximates arbitrary filters using various types of orthogonal polynomials, including Bernstein [22], Chebyshev [21], and Jacobi [74]. Another subclass parameterizes the filters by neural networks, including LanczosNet [40] and Specformer[3]. The robust modeling capability of spectral graph neural networks on data inspires us to leverage this foundation to tackle the issue of structural heterogeneity across domains.

### 2.2   Personalized Federated Learning

Federated learning [84, 34, 28, 14, 82] facilitates privacy-preserving collaborative learning on local data, introducing methods like FedAvg [51] to address this. Yet, it struggles with non-IID data across clients. Several techniques aim to address the challenge [33, 35, 30], but achieving a global model that suits all local data remains difficult [29]. Personalized Federated Learning (pFL) has attracted increasing attention for its ability to address the non-IID problem [13, 39, 60]. Research has approached improvements from various aspects. In personalized-aggregation-based methods, FedPHP aggregates the global model and old personalized models locally to preserve historical information [36], FedALA achieves personalized aggregation through personalized masks [91], and APPLE uploads only core models and employs directed relationship vectors for downloading [50]. In model-splitting-based approaches, FedRoD [6] learns with a global feature extractor and two heads for both global and personalized tasks. FedCP decouples features suitable for global and local heads through a conditional policy scheme [92]. Moreover, methods based on regularization and knowledge distillation have also been utilized to enhance pFL. However, pFL methods lack targeted strategy designs for graphs, making them not particularly suited for pFGL scenarios.

### 2.3   Federated Graph Learning

Recent studies have utilized the FL framework for distributed training of GNNs without sharing graph data [20, 41, 28, 9]. Current Federated Graph Learning (FGL) research can be categorized into two types: intra-graph and inter-graph FGL. In inter-graph FGL, each client has distinct graphs, and they jointly participate in federated learning to either improve GNN modeling of local data or achieve a model that can generalize across different datasets [77, 61]. Intra-graph FGL, on the other hand, aims to deal with challenges such as missing link prediction [11], subgraph community detection [93, 1], and node classification [25, 37]. However, most FGL methods lack specific design considerations for our scenario. More precisely, there is a general absence of consideration for the heterogeneity of graph-level structures and the personalized needs of different clients brought about by structural characteristics. In this paper, we focus on inter-graph FGL, taking into account spectral domain biases and the uniqueness of graph structures that result in client-specific preferences, to customize a personalized optimal model for each client specifically for graph classification tasks.

# 3 Preliminary

## 3.1 Graph Signal Filter

Assume that we have a graph $\mathcal{G} = (\mathcal{V}, \mathcal{E})$, where $\mathcal{V}$ represents the node set with $|\mathcal{V}| = n$ and $\mathcal{E}$ is the edge set. The corresponding adjacency matrix is defined as $A \in \{0, 1\}^{n \times n}$, where $A_{ab} = 1$ if there is an edge between nodes $a$ and $b$, and $A_{ab} = 0$ otherwise. The normalized graph Laplacian matrix is defined as $\tilde{L} = I_n - D^{-1/2} A D^{-1/2}$, where $I_n$ denotes the $n \times n$ identity matrix and $D$ is the diagonal degree matrix. We assume $\mathcal{G}$ is undirected. $\tilde{L}$ is a real symmetric matrix, whose spectral decomposition can be written as $\tilde{L} = U \Lambda U^T$, where the columns of $U$ are the eigenvectors and $\Lambda = \mathrm{diag}(\lambda_1, \lambda_2, \ldots, \lambda_n)$ are the corresponding eigenvalues ranged in $[0, 2]$. The Graph Fourier transform of a signal $x \in \mathbb{R}^{n \times d}$ is defined as $\tilde{x} = U^T x \in \mathbb{R}^{n \times d}$. The inverse transform is defined as $x = U\tilde{x}$ [57]. The $i$-th column of $U$ denotes a frequency component corresponding to the eigenvalue $\lambda_i$. Let $\tilde{x}_\lambda = U_\lambda^T x$, where $U_\lambda$ represents the eigenvector corresponding to $\lambda$, be the frequency component of $x$ at $\lambda$ frequency. We can use a function $g : [0, 2] \to \mathbb{R}$ to filter each frequency component by multiplying $g(\lambda)$. By defining $\Lambda = diag(\lambda)$, $g$ implements filtering on $\Lambda$, thus ultimately implementing filtering on the graph signal $x$. The whole process is defined as follows:

$$U g(\Lambda) U^T x. \tag{1}$$

By defining $g(\tilde{L}) = \sum_{k=0}^{K} \alpha_k \tilde{L}^k$, where $g$ is often set to be a polynomial of degree $K$ for parameterizing the filter, the filtering process can be rewritten as follows:

$$U g(\Lambda) U^T x = g(\tilde{L}) x. \tag{2}$$

## 3.2 Federated Learning and Personalized Federated Learning

Traditional FL leverages isolated data of distributed clients and collaboratively learns models $\mathcal{M}$ for a generalizable global model without leaking privacy. Specifically, the goal is to minimize:

$$\min_\theta f_g(\theta) = \min_\theta \sum_{i=1}^{N} w_i \mathcal{M}_i(\theta), \tag{3}$$

where $f_g(\cdot)$ denotes the global objective. It is computed as the weighted sum of the $N$ local objectives, with $N$ being the number of clients and $w_i \geq 0$ being the weights. The local objective $\mathcal{M}_i(\cdot)$ is often defined as the expected error over all data under local data $\mathcal{D}_i$.

In the context of personalized federated learning, the global objective takes a more flexible form:

$$\min_\Theta f_p(\Theta) = \min_{\theta_i, i \in [N]} \sum_{i=1}^{N} w_i \mathcal{M}_i(\theta_i), \tag{4}$$

where $f_p(\cdot)$ is the global objective for the personalized algorithms, and $\Theta = [\theta_1, \theta_2, \ldots, \theta_N]$ is the matrix with all personalized models. In this work, we aim to obtain the optimal $\Theta^* = \arg\min_\Theta f_p(\Theta)$, which equivalently represents the set of optimal personalized models $\theta_i, i \in [N]$.

# 4 Methodology

## 4.1 Generic Spectral Knowledge Sharing (GSKS)

**Motivation**. Current methods suffer from knowledge conflict arising from non-generic sharing under domain structural shifts. Since structural shifts impede the direct generic sharing at the structural level, we are the first to reveal and resolve knowledge conflicts from the spectral perspective. To explicitly address the spectral biases that reflect structural shifts in Fig. 1, we base our pFGL strategy on spectral GNNs and further propose Generic Spectral Knowledge Sharing (GSKS). Effective collaboration that overcomes spectral bias and structural shift is achieved, thereby addressing knowledge conflict. Details of GSKS are presented in Fig. 2 (a).

**Eigenvalue filtering**. Aiming at more expressive representations of frequency information, the eigenvalues are firstly mapped from scalars to meaningful vectors for subsequent learning of frequency

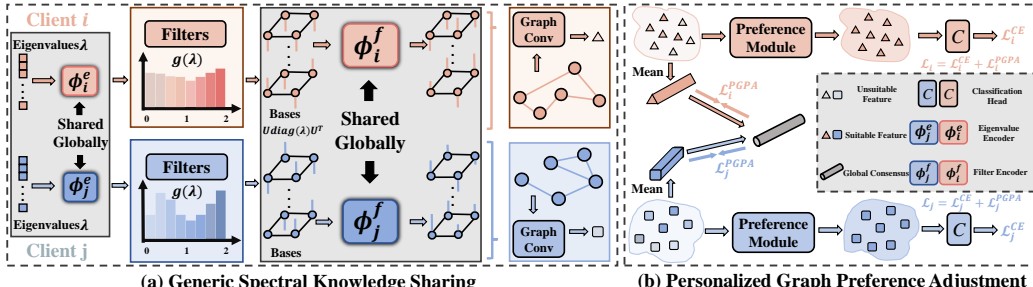

**(a) Generic Spectral Knowledge Sharing**   **(b) Personalized Graph Preference Adjustment**

Figure 2: **Architecture illustration** of FedSSP. The left box (a) refers to Generic Spectral Knowledge Sharing (GSKS), where we address knowledge conflict and promote effective global collaboration by **sharing generic** spectral knowledge extracted from spectral encoders $\phi^e$ and $\phi^f$ while **retaining non-generic** in other components. The right box (b) represents Personalized Graph Preference Adjustment (PGPA), where we leverage **preference module** guided by $\mathcal{L}_i^{PGPA}$ for satisfying inconsistent preferences and achieving suitable feature of datasets locally. These two boxes correspondingly refer to the two core strategies of our framework FedSSP.

interrelation in the multi-head attention module as follows:

$$\phi^e(\theta^e; \lambda) = \begin{cases} \sin(\beta\lambda/c^{q/d}), & \text{if } q \text{ is even,} \\ \cos(\beta\lambda/c^{(q-1)/d}), & \text{if } q \text{ is odd,} \end{cases} \tag{5}$$

where $c$ keeps eigenvalues within a suitable numerical range to distinguish different eigenvalues for trigonometric functions. $q \in \mathbb{Z} \cap [0, d-1]$ is the index for dimension $d$ while $\beta$ controlling the importance of $\lambda$ with defaulted value 10000. Moreover, $\theta^e$ denotes parameters of the eigenvalue encoder $\phi^e$, by which eigenvalues are mapped from scalars to vectors for richer frequency information. Consequently, they are more expressive for filtering by the attention module and decoder through $\mathbb{R}^1 \to \mathbb{R}^d$. The initial representations are the concatenation of eigenvalues and their encodings: $\lambda' = [concat[\lambda_1, \phi^e(\theta^e; \lambda_1)], \ldots, concat[\lambda_n, \phi^e(\theta^e; \lambda_n)]]^T \in \mathbb{R}^{n \times (d+1)}$. Then the multi-head attention module is leveraged. After stacking multiple transformer blocks, spectral decoder $\psi^d$ for eigenvalue decoding can learn new eigenvalues from the expressive representations of spectra:

$$\lambda_m = \psi^d(\text{Attention}(Q\theta_m^Q, K\theta_m^K, V\theta_m^V)), \tag{6}$$

Where the representations learned by the $m$-th head are fed into $\psi^d$, while $\psi^d$ denotes a combination of liner and optional activation. $\lambda_m \in \mathbb{R}^{n \times 1}$ is the filtered eigenvalues by the $m$-th head. The whole process in Eq. (6) acts as a spectral filter $g$ for the origin eigenvalues in Eq. (1).

To address the challenge of knowledge conflict, we attempt to explore the functionality of each module. The eigenvalue encoder $\phi^e$ captures multi-scale representations of eigenvalues and provides meaningful vectors of distinct frequencies. Since the mapping from eigenvalues to vectors by $\phi^e$ is independent of the domain characteristics, $\theta^e$ of $\phi^e$ is considered to contain generic knowledge. In contrast, as the spectral biases we reveal in Figure 1 demonstrate, biases exist in eigenvalue distribution across domains. In contrast, spectral characteristics within the same domain are more similar. Therefore, the attention module learns the non-generic magnitudes and relative dependencies specific to the spectral characteristics of each client. Correspondingly, the eigenvalue decoder focuses on decoding the most suitable message-passing scheme and client-specific frequency components from the representation processed by the attention module. Attention module and decoder together formed $g$ in Eq. (1), aiming at designing personalized filtered eigenvalue that guides message-passing at a personalized suitable frequency. Therefore, $\theta^e$ is shared in our strategy to achieve generic spectral knowledge sharing and effective collaboration unaffected by knowledge conflict.

Specifically, client $i$ uploads its update of $\theta_i^e$. At the $t$-th iteration ($t \geq 0$), the central server distributes global spectral encoder weights $\theta_g^t$. Accordingly, client $i$ updates local GNN weights including $\theta_i^e$ with their dataset $\mathcal{D}_i$ and send these updates as $\Delta\theta_i^t = \theta_i^t - \theta_g^t$ to the central server. Then the server aggregates the received local updates and modifies the global weight $\theta_g^{t+1}$ as follows:

$$\theta_g^{t+1} = \theta_g^t + \frac{\sum_{i=1}^N \Delta\theta_i^t}{N} (i \in [1..N]), \tag{7}$$

notably, aggregation based on sample size is unsuitable in this scenario for effective collaboration across various domains and datasets. Since clients here possess specific datasets with significant

quantitative variance, those with larger datasets tend to dominate the collaboration. Thereby preventing them from benefiting from the spectral and frequency knowledge of clients with fewer samples. Correspondingly, clients with fewer graphs are almost entirely dominated by knowledge that does not originate from their local data. To address the problem, we leverage a direct average of spectral encoder weights from all clients to achieve fair collaboration and cross-dataset knowledge sharing.

**Personalized graph convolution constructing.** After getting $M$ filtered eigenvalues, filter encoder: $\phi^f(\theta^f; B) \, \mathbb{R}^{M+1} \rightarrow \mathbb{R}^d$ is leveraged to construct the bases for personalized graph convolution. To avoid confusion and distinguish from the mentioned filter $g$ on eigenvalues in Eq. (1), filter here in filter encoder refers to the filtering on feature message-passing through bases in graph convolution. New bases are first reconstructed and concatenated along the channel dimension. Specifically, by defining $\Lambda_m = diag(\lambda_m)$, they are fed into filter encoder $\phi^f$ as follows:

$$B_m = \mathbf{U}\Lambda_m\mathbf{U}^T, \quad \forall m \in \{1, \ldots, M\}, \tag{8}$$

$$\hat{B} = \phi^f(\theta^f; B), \tag{9}$$

where $B = [B_1, B_2, \ldots, B_M] \in \mathbb{R}^{n \times n \times M}$ while $B_m \in \mathbb{R}^{n \times n}$ is the $m$-th new basis and $\hat{B} \in \mathbb{R}^{n \times d}$ is for feature filtering in graph convolution ultimately. The original bases $B_m$ are initially constructed from the eigenvectors $U$ and the filtered eigenvalues $\lambda_m$, with $\phi^f(\theta^f; B)$ responsible for the learnable transformation of bases from original to new. This transformation facilitates learning more suitable schemes for graph message-passing and processing at various frequencies. Filter encoders $\phi^f(\theta^f; B)$ in clients encapsulate knowledge of various frequency components which affects how much graph signal varies from nodes to their neighbors for better graph convolution construction. Due to restrictions on the sample size for certain clients, they are unable to adequately learn message-passing techniques for handling specific frequency components. As a solution, the filter encoder is shared among clients, enabling them to fully acquire the graph signal processing methods for frequencies that are challenging to learn locally. Specifically, collaboration on filter encoder can aid $\phi^f(\theta^f; B)$ of each client in learning how to construct suitable graph convolution from various message-passing schemes. Therefore, we design client $i$ to upload the weights $\theta_i^f$ of its $\phi_i^f$ the same way as $\phi_i^e$ Eq. (7), thereby achieving a comprehensive understanding of different frequency messages in graphs. Subsequently, it enables the construction of powerful personalized graph convolutions as follows:

$$x_v' = \sigma\left((\hat{B} \cdot x_v)\theta^{Conv}\right) + x_v, \tag{10}$$

where $x_v$ is the node $v$ representation from the previous layer, while $x_v'$ represents the output of the current layer. $\theta^{Conv}$ denotes weights of graph convolution, and $\sigma$ refers to the optional activation. Ultimately, the representations of all nodes within a graph are aggregated by an average pooling layer to form the overall feature representation of graph $\mathcal{G}_l$ in dataset $\mathcal{D}_i$ of client $i$ as follows:

$$h_l = \frac{1}{|\mathcal{V}|} \sum_{v=1}^{|\mathcal{V}|} x_v, \quad \forall l \in \{1, \ldots, |\mathcal{D}_i|\}, \tag{11}$$

where $h_l$ is defined as the average of all node features in graph $\mathcal{G}_l$, namely the graph feature, while $\mathcal{V}$ refers to the node quantities in graph $l$ here. By sharing generic spectral knowledge and retaining client-specific knowledge we successfully achieve effective collaboration that overcomes spectral bias, thereby domain structural shift from the spectral perspective.

## 4.2 Personalized Graph Preference Adjustment (PGPA)

**Motivation.** Due to the GNN message-passing nature, distinct graph structures prefer different message-passing schemes, especially when meeting the specificity of datasets in cross-dataset and cross-domain scenarios. Consequently, The spectral encoders under global collaboration fail to satisfy the inconsistent local preferences of graphs. Correspondingly, graph convolutions tend to learn biased message-passing schemes, thereby extracting unsuitable graph features. Our approach provides personalized adjustments to address this challenge based on client preference. Moreover, we solve the over-reliance issue that arises from the process of satisfying various preferences. Details of GSKS are presented in Fig. 2 (b).

To satisfy the various preferences and make the graph features more suitable for graphs, we propose a learnable preference module that adjusts to features extracted by client $i$ to satisfy local graph

structure preference explicitly. The module includes learnable parameters matched in size with the feature space, thus acting as a refined tool to flexibly satisfy the preferences of each client during local training. Considering local model $\mathcal{M}$ including feature extractor $\mathcal{F}(\theta^F; \mathcal{G})$, classification head $\mathcal{C}(\theta^C, h)$, and preference module $\mathcal{P}(\delta)$, where $\mathcal{G}$ represents graphs contained in dataset $\mathcal{D}$ of a client. The whole graph feature-extracting process can be defined as follows in our strategy:

$$h = \mathcal{F}(\theta^F; \mathcal{G}), \quad h' := h + \delta, \tag{12}$$

by integrating the original feature $h$ with preference adjustments $\delta$, $h'$ becomes the ultimately suitable feature that satisfies client preference. Now we leverage adjusted feature $h'$ for $z' = \mathcal{C}(\theta^C; h')$ instead of the original unsuitable representation $h$ . Specifically, the local loss for client $i$ is:

$$\mathcal{L}_i = \mathbb{E}_{(z'_i, y_i) \sim \mathcal{D}_i} \mathcal{L}_i^{CE} = \mathbb{E}_{(z'_i, y_i) \sim \mathcal{D}_i} CE(z'_i, y_i), \tag{13}$$

where the Cross-entropy (CE) loss measures the difference between the predicted probability and the true label. Nevertheless, the preference module learns not only the client-specific preference but also aspects that should be handled by the feature extractor $\mathcal{F}$ without a guide for preference. As a result, the local feature extractor $\mathcal{F}$ might overly rely on adjustments provided by the preference module during training, thereby hindering its capability. Correspondingly, this over-reliance can negatively impact federated collaboration. When the capability of $\mathcal{F}$ is degraded, the shared spectral encoders fail to convey beneficial knowledge to others, leading to unpromising collaboration.

Therefore, it is essential to guide the preference module to focus solely on the aspects of client preferences, rather than interfering with the feature extraction guided by collaboration. We achieve this by guiding the output of the feature extractor to align more closely with global graph features. Consequently, the PGPA module is directed to concentrate on client preferences. To implement this, we first calculate the mean of local graph features in each iteration:

$$\bar{h}_i = (1 - \mu) \cdot \bar{h}_i^{\text{pre}} + \mu \cdot \bar{h}_i^{\text{cur}}, \tag{14}$$

where $\mu$ denotes the momentum we introduce for bringing graph modeling patterns from previous batches to the current batch in the same local epoch. $\bar{h}_i^{\text{pre}}$ and $\bar{h}_i^{\text{cur}}$ represent the local mean graph feature of the previous batches and the current batch. It is necessary to distinguish between mean and prototype. In this scenario, clients own various datasets, thus the class information is client-specific. Correspondingly, the mean $h_i$ which represents the average modeling for graphs in client $i$ is unrelated to class information. After local training, client $i$ uploads its mean to the server for global consensus aggregation. Based on our exploration of Eq. (7), a direct average is leveraged here:

$$\bar{h}_g = \frac{\sum_{i=1}^{N} \bar{h}_i}{N}, \tag{15}$$

where $\bar{h}_g$ refers to the global graph modeling consensus calculated from all samples across all clients. Accordingly, we employ the Mean Squared Error (MSE) to measure the distance between the local graph feature mean and the global graph mean obtained from the previous round. This measurement serves as a regularization term to encourage the local graph feature to align closer to the global modeling consensus, thus guiding the preference module to focus on preference and correspondingly address the over-reliance issue. Specifically, local feature extractors are encouraged to extract certain frequency messages in graphs that reflect the global modeling consensus, making the PGPA only responsible for client-specific preference. The local loss of client $i$ is now defined as:

$$\mathcal{L}_i = \mathbb{E}_{(z'_i, y_i) \sim \mathcal{D}_i} (\mathcal{L}_i^{CE} + \mathcal{L}_i^{PGPA}) = \mathbb{E}_{(z'_i, y_i) \sim \mathcal{D}_i} [CE(z'_i, y_i) + \tau \cdot \text{MSE}(\bar{h}_i, \bar{h}_g)]. \tag{16}$$

By implementing $\text{MSE}(\bar{h}_i, \bar{h}_g)$, we explicitly align local graph modelings with the global consensus, thus guiding the preference module $\mathcal{P}$ to focus on preference and addressing the considered issue of over-reliance by forcing the preference module to focus on client-specific graph preference.

## 5 Experiments

### 5.1 Experimental Setup

We perform experiments on graph classification tasks in various cross-dataset and cross-domain scenarios to validate the superiority of our framework FedSSP.

**Datasets.** Follow the settings in [61], we use 15 public graph classification datasets from four different domains, including Small Molecules (MUTAG, BZR, COX2, DHFR, PTC_MR, AIDS,

NCI1), Bioinformatics (PROTEIN, OHSU, Peking_1), Social Networks(IMDB-BINARY, IMDB-MULTI), and Computer Vision (Letter-low, Letter-high, Letter-med) [52]. Node features are available in some datasets, and graph labels are either binary or multi-class. We create six different settings: (1) cross-dataset setting based on seven small molecules datasets (SM); (2)-(6) both cross-dataset and cross-domain settings based on datasets from two different domains(BIO-SM, SM-CV) and three different domains(BIO-SM-SN, BIO-SN-CV, CHEM-SN-CV)

**Baselines.** We compare ours with several SOTA federated approaches. (1)**Local** as the first baseline; (2)**FedAvg** [51]; (3)**FedProx** [35] which address heterogeneity issues in FL; (4)**APPLE** [50] and (5)**FedCP** [92], two state-of-the-art pFL method;(6)**FedSage** [93], (7)**GCFL** [77] ,(8)**FGSSL** [25], and (9)**FedStar** [61], four state-of-the-art FGL methods.

**Implementation Details.** The experiments are conducted using NVIDIA GeForce RTX 3090 GPUs as the hardware platform, coupled with Intel(R) Xeon(R) Gold 6240 CPU @ 2.60GHz. For each setting, every client holds its unique graph dataset, among which 10% are held out for testing, 10% for validation, and 80% for training. We leverage the AdamW optimizer [31] for local GNNs with learning rate 0.001, the default parameter of $\epsilon = 1e - 8$, and $(\beta_1, \beta_2) = (0.99, 0.999)$, as suggested by [54, 85]. The number of communication rounds is 200 for all FL methods. We report the results with an average of over 5 runs of different random seeds.

## 5.2 Experimental Results

**Performance Comparison** We show the federated graph classification results of all methods under six different non-IID settings, including one cross-dataset setting (SM), two cross-double-domain settings (BIO-SM, SM-CV) cross-multi-domain settings (BIO-SM-SN, BIO-SN-CV, SM-SN-CV). We summarize the final average test accuracy in Tab. 1. These results indicate that FedSSP outperforms all other baselines in five out of the six settings. Notably, traditional FL algorithms such as FedAvg and FedProx failed to outperform self-training due to the strong cross-datasets and cross-domain non-IID challenge of this scenario. Correspondingly, algorithms such as FedStar and FedCP which are designed specifically for pFGL or pFL scenarios perform better here.

Table 1: Comparison with the state-of-the-art methods on one cross-dataset and five cross-domain settings. Best in bold and second with underline. In each setting, each client owns a unique dataset.

| Methods | single-domain | double-domain | | Multi-Domain | | |
| --- | --- | --- | --- | --- | --- | --- |
| | SM | SM-BIO | SM-CV | SM-BIO-SN | BIO-SN-CV | SM-SN-CV |
| Local | $77.33 \pm 1.15$ | $72.52 \pm 1.86$ | $82.24 \pm 1.73$ | $71.13 \pm 1.32$ | $72.59 \pm 2.70$ | $77.83 \pm 0.54$ |
| FedAvg [ASTAT17] | $74.12 \pm 2.10$ | $67.82 \pm 1.63$ | $81.21 \pm 1.00$ | $67.31 \pm 2.56$ | $70.93 \pm 2.91$ | $75.33 \pm 1.06$ |
| FedProx [arXiv18] | $69.35 \pm 3.36$ | $67.27 \pm 4.17$ | $70.02 \pm 2.27$ | $63.89 \pm 4.33$ | $69.32 \pm 1.75$ | $67.15 \pm 2.25$ |
| FedSage [NeurIPS21] | $75.61 \pm 1.16$ | $72.60 \pm 3.18$ | $76.23 \pm 0.49$ | $70.84 \pm 0.88$ | $69.69 \pm 1.11$ | $73.36 \pm 0.86$ |
| GCFL [NeurIPS21] | $77.71 \pm 1.53$ | $72.05 \pm 2.20$ | $72.64 \pm 0.71$ | $70.43 \pm 1.39$ | $67.91 \pm 2.15$ | $71.79 \pm 0.21$ |
| APPLE [IJCAI22] | $74.29 \pm 1.89$ | $70.40 \pm 2.13$ | $76.07 \pm 2.55$ | $71.07 \pm 1.64$ | $72.52 \pm 1.03$ | $72.33 \pm 0.42$ |
| FedCP [KDD23] | $77.58 \pm 2.00$ | $71.15 \pm 1.77$ | $81.59 \pm 0.40$ | $71.32 \pm 1.23$ | $\underline{73.74 \pm 2.53}$ | $\underline{78.17 \pm 1.78}$ |
| FGSSL [IJCAI23] | $77.90 \pm 0.85$ | $72.47 \pm 2.15$ | $\underline{82.60 \pm 0.48}$ | $68.13 \pm 1.71$ | $73.44 \pm 1.33$ | $77.90 \pm 0.62$ |
| FedStar [AAAI23] | $78.63 \pm 2.11$ | $72.71 \pm 1.22$ | $78.84 \pm 1.07$ | $\mathbf{72.60 \pm 2.45}$ | $69.51 \pm 3.24$ | $75.94 \pm 0.40$ |
| **FedSSP (ours)** | $\mathbf{79.62 \pm 2.23}$ | $\mathbf{73.66 \pm 2.34}$ | $\mathbf{84.29 \pm 0.68}$ | $\underline{72.37 \pm 2.18}$ | $\mathbf{75.07 \pm 2.70}$ | $\mathbf{79.12 \pm 1.23}$ |

**Convergence Analysis** Fig. 3 shows the curves of the average test accuracy with standard deviation during the training process across five random runs of three settings (SM, SM-CV, SM-SN-CV) representing single-domain, double-domain, and multi-domain scenarios, including the results of various baselines. As is shown, traditional FL methods such as FedAvg or FedProx own higher standard deviations and are more unstable while methods designed specifically for pFGL scenarios such as GCFL and FedStar are more stable with lower standard deviations.

## 5.3 Ablation Study

**Effects of Key Components Mechanism of FedSSP** To better understand the impact of specific design components on the overall performance of FedSSP, we conducted an ablation study in which

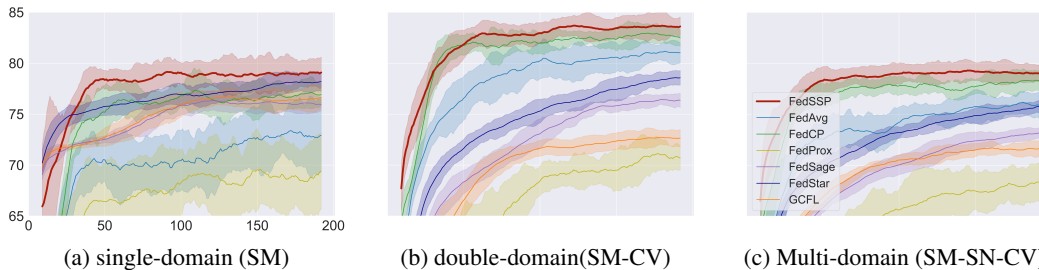

|  |  |  |
|---|---|---|
| (a) single-domain (SM) | (b) double-domain(SM-CV) | (c) Multi-domain (SM-SN-CV) |

Figure 3: Test accuracy curves of FedSSP and six other methods along the communication rounds on our three different settings(SM, SM-CV, SM-SN-CV). The y-axis range is from 65 to 85 for all settings.

we varied these components of single, double, and multi-domain settings(SM, SM-CV, SM-SN-CV). As shown in Tab. 2, compared to FedAvg, GSKS significantly enhances accuracy when applied independently. Correspondingly, as a further exploration of the nature of GNN message passing, PGPA achieves noticeable success in adjusting client preferences.

**Effects of Key Component Mechanism of GSKS** Tab. 3 discuss the key component of GSKS. We demonstrated the impact of different sharing strategies. Specifically, sharing only non-generic spectral GNN components or all components often fails to outperform self-training, while GSKS successfully dominates all the strategies. Accordingly, the results fully validate the effectiveness of GSKS in sharing universal knowledge and promoting effective collaboration in this scenario.

Table 2: **Ablation study** of key components of FedSSP on single-domain, double-domain, and multi-domain settings (SM, SM-CV, SM-SN-CV).

| GSKS | PGPA | Setting | | |
|---|---|---|---|---|
| | | SM | SM-CV | SM-SN-CV |
| ✗ | ✗ | 74.12 | 81.21 | 75.33 |
| ✓ | ✗ | 77.83 | 82.78 | 78.54 |
| ✗ | ✓ | 74.59 | 81.33 | 76.12 |
| ✓ | ✓ | **79.62** | **84.29** | **79.12** |

Table 3: **Ablation study** of key components of GSKS on a single-domain, double-domain, and multi-domain settings (SM, SM-CV, SM-SN-CV).

| Ours | Other | Setting | | |
|---|---|---|---|---|
| | | SM | SM-CV | SM-SN-CV |
| ✗ | ✗ | 77.33 | 82.24 | 77.83 |
| ✓ | ✓ | 74.12 | 81.21 | 75.33 |
| ✗ | ✓ | 77.21 | 81.64 | 78.17 |
| ✓ | ✗ | **77.83** | **82.78** | **78.54** |

## 5.4 Hyper-parameter Study

We compare the graph classification performance under different values of PGPA parameter $\tau$, momentum $\mu$, number of attention heads, and hidden dimension. Where Fig. 4 shows the results when these hyper-parameters are fixed at different scales and values. It indicates that the choosing of $\tau$ can affect the strength of PGPA while performance is not influenced much unless they are set to extreme values. All studies of $\tau$ and $\mu$ here outperform the baseline. We also test the performances under different values of attention heads and hidden dimensions. For results in Tab. 1, we set up 4 heads for the multi-head attention mechanism while 128 for the hidden dimension.

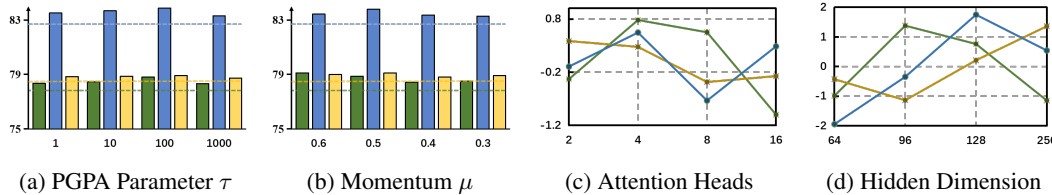

|  |  |  |  |
|---|---|---|---|
| (a) PGPA Parameter $\tau$ | (b) Momentum $\mu$ | (c) Attention Heads | (d) Hidden Dimension |

Figure 4: **Analysis on hyper-parameter in FedSSP**. Graph classification results under different $\tau$, $\mu$, attention heads, and hidden dimensions. Colors green, blue, and yellow refer to performance on single, double, and multi-domain settings (SM, SM-CV, SM-SN-CV). The dashed lines of corresponding colors represent the baseline test accuracy for each setting, which includes only the GSKS strategy.

# 6 Discussion

Even though FedSSP has achieved significant success in cross-domain federated graph learning collaborations, it still faces certain limitations as a spectral GNN-based approach. Compared to spatial GNNs, while spectral GNNs have the advantage of overcoming structural heterogeneity from the spectral domain, many spectral GNNs require eigenvectors and eigenvalues, which adds to the computational overhead of data preprocessing and subsequent storage burden.

Furthermore, we notice a similar approach in DBE [90] which employs static global consensus in FL to separate personalized and global information. Nevertheless, it inevitably struggles to handle scenarios where the message-passing of multiple GNNs is continuously updated. It merely provides a static anchor point, making it difficult to establish a global graph modeling consensus that could guide the local GNNs in capturing graph signals. Instead, we align the GNN backbone with dynamic global graph modeling to avoid the preference module from overly extracting features that should be captured by the GNN itself, which could lead to decreased GNN performance and hinder global collaboration. This approach allows for real-time adjustment of message-passing across different client GNNs, focusing the preference module solely on personalization. Additionally, to address issues such as sample size disparity between domains and dominance of large domains in model parameter aggregation, we adopt a direct averaging strategy in dynamic global aggregation instead of conventional weighted averaging to mitigate these concerns.

# 7 Conclusion

In this paper, we pioneer an innovative exploration of cross-domain personalized Federated Graph Learning. To achieve this goal, we achieve improvements in two aspects: seeking effective global collaboration and suitable local application, thus proposing a novel framework FedSSP. For global collaboration, GSKS is leveraged to facilitate the sharing of generic spectral knowledge, overcoming knowledge conflict by domain structural shift from a spectral perspective. For local applications, we design PGPA to satisfy inconsistent preferences of specific datasets contained in clients. By integrating these two strategies, FedSSP outperforms various state-of-the-art methods on various cross-dataset and cross-domain pFGL scenarios in graph classification.

# 8 Acknowledgment

This work is supported by National Natural Science Foundation of China under Grant (62361166629, 62176188, 623B2080), Key Research and Development Project of Hubei Province (2022BAD175), and the Luojia Undergraduate Innovation Research Fund Project of Wuhan University. The numerical calculations in this paper have been supported by the super-computing system in the Supercomputing Center of Wuhan University.

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
