# OpenReview forum: "FedSSP: Federated Graph Learning with Spectral Knowledge and Personalized Preference"
_NeurIPS.cc/2024/Conference — NeurIPS 2024 poster_

### Official Review · Reviewer_cuMU · 2024-07-08

**Soundness:** 3
**Presentation:** 3
**Contribution:** 4
**Rating:** 7
**Confidence:** 4

**Summary:**

This paper focuses on cross-domain Federated Graph Learning, where graph data stored in clients exists negative domain structural shifts. Authors observe the presence of spectral biases as a reflection of structural shifts. Thereafter, this work proposes the Generic Spectral Knowledge Sharing (GSKS), which allows clients to share certain components containing generic spectral knowledge. Moreover, it proposes Personalized Graph Preference Adjustment (PGPA) to satisfy preferences of clients for more suitable local applications. Extensive experiments on different scenarios demonstrates the superiority of the proposed methods.

**Strengths:**

1. Well-motivated. In particular, it first observes the spectral biases as a reflection of structural shifts and further provides a targeted solution to address the biases. Moreover, the authors acknowledge the unique preferences of the dataset in this scenario may lead to the global message passing being not fully applicable to local data and accordingly propose appropriate solutions for the particularity of datasets.

2. Very easy to follow. The Figure is well presented. In particular, the authors provide a well-defined motivation visualization, helping with the understanding of specific challenges and the framework details. In addition, the precise framework presentation aids in the comprehension of each module’s role.

3. Structural heterogeneity is a common challenge in federated graph learning. The innovative solution presented in this paper offers a new perspective for future advancement by promoting improvements in spectral GNNs instead of traditional ones. It is not necessarily limited to cross-domain scenarios but can also bring new ideas for addressing structural heterogeneity in other non-IID federated graph learning contexts.

**Weaknesses:**

1. Lacking of comprehensive comparative analysis of existing FGL methods. For instance, FGSSL appears in the performance table but is not comparatively analyzed in the introduction. The authors are supposed to clarify the shortcomings of this method in the given scenario.

**Questions:**

I expected a more detailed comparative analysis of FGL baselines mentioned in the weakness. Could the authors provide a discussion on it?

**Limitations:**

This work solves the problems in cross-domain settings well. I wonder whether it is generalizable enough to be extended to other non-iid problems, such as each client having a portion of the same graph dataset. It’s necessary to add relevant experiments to demonstrate that this method is not limited.

---

> ### Author Rebuttal · Authors · 2024-08-04
>
> Dear Reviewer cuMU:
>
> Thank you for your thorough review and the kind words about our well-motivated study and handling of structural heterogeneity. We sincerely appreciate your time and effort. We hope that our responses below will address your concerns.
>
> ### Weakness
>
> **W1: Lacking comprehensive comparative analysis of existing FGL method FGSSL.**
>
> A1: In scenarios with strong structural heterogeneity across datasets and domains, alignment to global structural knowledge in FGSSL is inevitably negatively impacted. Even with adjustments to align client neighbor modeling knowledge to the global GNN, it remains challenging to aggregate a highly generalizable global model under domain structural bias. Furthermore, even though its node-level strategies are beneficial for better graph modeling, the effectiveness of the method decreases under the influence of structure-biased message passing. Specifically, clients are inevitably influenced by the message-passing schemes of GNNs trained on graph data with significantly different domain structures, thus inevitably failing to model their local graph data accurately.
>
>
> ### Limitations
>
> **Uncertainty about generalizability to other non-iid problems**
>
> A2: Theoretically, our method can be extended to scenarios where each client possesses a portion of a dataset and different clients may have different datasets. Therefore, we validated the scalability of our method in this scenario and conducted the following experiments for demonstration.
>
> Specifically, we conduct experiments where each of the seven small molecule datasets is split into 1-11 segments. Different from cross-domain and cross-dataset settings in this paper, it simulates non-iid scenarios where different clients may have portions of the same dataset. It denotes that FedSSP consistently outperformed the other methods across all scenarios.
>
> *Table: **Extended non-iid scenarios** where each client possesses a portion of a small molecule dataset*
>
> | Client Number | 7    | 21   | 35   | 49   | 63   | 77   |
> |:---:| :--:|:---:|:---:|:---:|:---:|:---:|
> | FedAvg       | 74.12 | 71.92 | 70.75 | 71.16 | 73.66 | 74.03 |
> | FedStar   | 78.63 | 76.37 | 74.18 | 73.20 | 75.81 | 78.28 |
> | FedSSP     | **79.62** | **76.74** | **74.86** | **74.31** | **76.25** | **79.44** |

---

> ### Author Response · Authors · 2024-08-11
>
> Dear Reviewer cuMU,
>
> We highly value the time and effort you have dedicated to evaluating our work. We are fully aware of the importance of your time and strive to respect it. In light of this, we would greatly appreciate any additional feedback or confirmation that our rebuttal has effectively addressed your comments. Our goal is to ensure that we have comprehensively addressed your concerns.
>
> Thank you so much for your time and consideration.
>
> Authors

---

### Official Review · Reviewer_3kee · 2024-07-09

**Soundness:** 3
**Presentation:** 3
**Contribution:** 3
**Rating:** 6
**Confidence:** 3

**Summary:**

The paper introduces a novel method FedSSP designed to address the current limitations of personalized Federated Graph Learning methods. The author highlighten that existing methods fail to deal with domain structural shift and ignore the uniqueness of datasets in cross-dataset scenarios. To address the limitations, FedSSP innovatively utilizes spectral GNNs and shares key layers in them to facilitate generic communication. Meanwhile, the authors propose preference module for the preferences derive from different graph datasets.

**Strengths:**

1. Addressing problems in pFGL from a spectral perspective is novel and interesting, setting a new direction for future research in federated graph learning.
2. The motivation is clear and compelling. The sharing strategy tackles the issue of knowledge conflict and spectral bias. Preference adjustment is employed to cater to the structural characteristics and preferences of individual datasets, thus ensuring that the graph features extracted are highly relevant and suitable.
3. The experiment is comprehensive, covering a wide range of scenarios to thoroughly evaluate the proposed methods. The ablation studies are thoroughly conducted, providing clear insights into the contributions of each component and demonstrating its effectiveness.

**Weaknesses:**

1. Since the method of spectral encoder sharing essentially belongs to the personalized layer methods, I believe the authors need to discuss the differences between this method and other personalized layer methods in this paper, which they did not, such as APPLE.
2. How the node-level method FGSSL is used here in the experiments? What is the core model of APPLE here in the federated graph learning scenario? Could the authors provide the implementation details of the baselines?
3. The author changed the model architecture here but has not compared its communication cost with baselines.

**Questions:**

Please refer to the Weakness part.

**Limitations:**

Authors should discuss communication cost for this potential limitation.

---

> ### Author Rebuttal · Authors · 2024-08-04
>
> Dear Reviewer 3kee:
>
> We deeply appreciate your positive feedback regarding the innovative aspects of our methods and the comprehensiveness of our experiments. Thank you for your time and effort in reviewing our paper. We hope that our responses below will address your concerns and further affirm the quality of our research.
>
> ### Weakness
>
> **W1: Insufficient discussion on differences between spectral encoder sharing and other personalized layer methods like APPLE.**
>
> A1: Following the implementation of the personalized layer method FedPer in previous FGL research, the implementation of APPLE here shares the graph convolution parameters in GNNs. Due to the lack of targeted solutions for cross-dataset and cross-domain scenarios, direct sharing of all graph convolution parameters inevitably struggles with structural conflict.
>
> Instead, collaboration under our proposed solution overcomes the conflict from the spectral perspective. Based on our investigation of domain spectral biases, we achieved effective cross-domain collaboration by exploring components unaffected by structural bias in spectral GNN.
>
> **W2: Lack of implementation details for node-level method FGSSL and core model APPLE in federated graph learning.**
>
> A2: FGSSL primarily includes two strategies: federated graph structure distillation (FGSD) to enhance node feature modeling, and federated node semantic contrast (FNSC) to encourage the neighborhood modeling and structural processing of each client to approach the global level.
>
> The node features before GNN average pooling is leveraged for similarity matrix calculation in FGSD. Specifically, in our graph-level task, each client possesses multiple graphs, each graph having its feature similarity matrix. Based on this, they converge towards the inherent adjacency relationships provided by the global model. For FNSC, since there are no unified node labels across clients in the scenario, we implemented contrastive learning using an unsupervised strategy to replicate feature optimization of FGSSL in graph-level scenarios.
>
> In addition, the core model part in APPLE here refers to the graph convolution parameters as we illustrated in A1.
>
> **W3: No comparison of communication cost with baselines.**
>
> A3: Since our strategy only shares knowledge of the eigenvalue encoder and filter encoder, it is certain that our communication cost is significantly lower than all baselines. Whether it is the traditional approach of sharing all parameters, the sharing of structure processing channels in FedStar, or the sharing of graph convolution parameters in APPLE, our communication cost is lower. If accepted, our final version will include a discussion on communication costs.

---

> > ### Comment · Reviewer_3kee · 2024-08-11
> >
> > Thank you for the detailed response. I  maintain my score.

---

> ### Author Response · Authors · 2024-08-11
>
> Dear Reviewer 3kee,
>
> We would like to express our heartfelt thanks for your consistent support. Your positive assessment of our work is greatly appreciated. Please feel free to reach out if you have any additional thoughts or suggestions.
>
> Thank you for your consideration and valuable time.
>
> Authors

---

### Official Review · Reviewer_jSYo · 2024-07-10

**Soundness:** 3
**Presentation:** 3
**Contribution:** 2
**Rating:** 6
**Confidence:** 3

**Summary:**

This work proposed a personalized federated graph learning framework for federated graph classification. The framework includes strategies for sharing generic knowledge and satisfying personalized preferences. The authors evaluated six different cross-dataset and cross-domain settings and showed good performance.

**Strengths:**

1. The writing in this paper is coherent, making the proposed concepts and methodologies easy to follow. The logical flow and concise explanations enhance overall readability and comprehension.
2. The methods are thoughtfully designed with a deep understanding of the specific challenges inherent in cross-domain scenarios. They effectively address these targeted issues, ensuring good performance and adaptability across diverse datasets and domains.

**Weaknesses:**

1. In the motivation, the authors do not clearly explain the relationship between the two spectral bias metrics and the graph structure.
2. The scalability of the method is not well analyzed in the experiment.
3. Absence of a detailed discussion on the necessity of the shared filter encoder in the proposed method, particularly in relation to its specific architecture.

**Questions:**

Q1) I have some confusion regarding the two spectral bias metrics used in the motivation. Could the authors specifically point out the relationship between these two spectral bias metrics and graph structure?
Q2) Could the authors add experiments on the scalability of the method, such as performance with varying numbers of clients?
Q3) The global consensus in the method is similar to the prototype. If it differs from prototype, authors should provide distinctions and explanations.

**Limitations:**

Yes

---

> ### Author Rebuttal · Authors · 2024-08-04
>
> Dear Reviewer jSYo:
>
> We sincerely appreciate your time and effort in reviewing our paper, and we are grateful for your positive feedback on our writing and the effectiveness of our methodology. We hope that our responses below will address your concerns and lead to an updated score.
>
> ### Weakness
>
> **W1 & Q1: How spectral bias metrics are related to graph structure?**
>
> A1: Eigenvalues and the Fiedler value are significantly associated with the graph structure. In the first place, eigenvalues and eigenvectors can be employed for graph clustering and partitioning. For instance, the k smallest non-zero eigenvalues and their corresponding eigenvectors can be utilized for k-means clustering of the graph. Furthermore, the Fiedler value can assess the connectivity of the graph [1]. The Fiedler value and its corresponding eigenvector, known as the Fiedler vector, also play a crucial role in community identification [2].
>
> **W2 & Q2: Validation of the scalability of the proposed method under varying numbers of clients.**
>
> A2: Theoretically, our method can be extended to scenarios where each client possesses a portion of a dataset and different clients may have different datasets. Therefore, we validated the scalability of our method based on these scenarios.
>
> We conducted experiments in SM cross-dataset settings under varying client scales. Specifically, we performed experiments with the client scales ranging from 7 to 77 where each of the seven small molecule datasets was split into 1-11 segments. The results in the following Table denote that FedSSP consistently outperforms the other methods across all scenarios.
>
> *Table: **Performance under varying client scales***
>
> | Client Number | 7    | 21   | 35   | 49   | 63   | 77   |
> |:---:| :--:|:---:|:---:|:---:|:---:|:---:|
> | FedAvg       | 74.12 | 71.92 | 70.75 | 71.16 | 73.66 | 74.03 |
> | FedStar   | 78.63 | 76.37 | 74.18 | 73.20 | 75.81 | 78.28 |
> | FedSSP     | **79.62** | **76.74** | **74.86** | **74.31** | **76.25** | **79.44** |
>
> **W3: Absence of a detailed discussion on the necessity and architecture of the shared filter encoder.**
>
> A3: Filter encoders encapsulate knowledge of various frequency components which affects how much graph signal varies from nodes to their neighbors for better graph convolution construction. This strategy benefits clients by enabling them to acquire knowledge of transmitting graph signals at different frequencies from other GNNs, thereby promoting the construction of graph convolutions for individual clients. Furthermore, the functionality is discussed in detail on line 253 of our paper.
>
> ### Questions
>
> **Q3: How does the prototype differ from global consensus?**
>
> A4: Our paper focuses on federated graph learning in cross-dataset and cross-domain scenarios, which denotes that each client possesses its unique class information, thus rendering the concept of prototypes inapplicable here. The global consensus in this paper is derived from the feature mean of graphs across all clients in each round, without the class information contained in prototypes. Global consensus represents the knowledge for graph data modeling by all participating GNNs and provides standardized global graph processing knowledge, which the prototypes cannot achieve in this scenario. In addition, this difference is discussed in detail on line 253 of our paper. We hope this provides the necessary clarification.
>
> [1] Fiedler M. Algebraic connectivity of graphs[J]. Czechoslovak mathematical journal, 1973, 23(2): 298-305.
>
> [2] Von Luxburg U. A tutorial on spectral clustering[J]. Statistics and computing, 2007, 17: 395-416.

---

> > ### Comment · Reviewer_jSYo · 2024-08-11
> >
> > Thanks for the rebuttal. After reading other reviewers' comments, I decided to raise my original score.

---

> ### Author Response · Authors · 2024-08-11
>
> Dear Reviewer jSYo,
>
> Thank you very much for your prompt response and for reconsidering your evaluation. We are truly grateful for your valuable feedback and the time you have invested in our manuscript. Your comments have greatly contributed to improving our work, and we appreciate your support in advancing it through this process.
>
> If there is anything further you believe could be refined or enhanced, please let us know.
>
> Authors

---

### Official Review · Reviewer_LD7P · 2024-07-11

**Soundness:** 3
**Presentation:** 4
**Contribution:** 3
**Rating:** 7
**Confidence:** 4

**Summary:**

FedSSP tackles structural heterogeneity well in personalized Federated Graph Learning. When it comes to cross-domain scenarios, the structural heterogeneity becomes more negative than usual. It is crucial to mitigate the impact of domain structural shifts. FedSSP proposes two strategies to address these challenges from two directions. It firstly overcomes knowledge conflict by generic spectral encoder weights to seek better collaboration across various datasets. Besides, considering the uniqueness of clients in this scenario, it then satisfies the special preferences of the graphs by feature adjusting.

**Strengths:**

- Studies on cross-domain federated graph learning are crucial for the applications of federated learning in the real world. This would help increase the generalizability in practical applications. This study promotes the generalizability of federated graph learning and demonstrates superior performance in various simulated settings, proving its advantage and broad applicability in practical applications.
- The authors present a novel perspective and a spectra-based solution to tackle the issue of structural heterogeneity among clients. The proposed approach effectively addresses the variations in graph structures across clients, thereby enhancing the overall coherence and performance of the federated learning framework.
- The generic spectral knowledge sharing and preference modules work harmoniously and complement each other effectively. After addressing knowledge conflicts through spectral bias mitigation, personalized adjustments act as further personalized optimizations, enhancing generic message passing. They together enable adaptive collaboration across diverse datasets and domains.

**Weaknesses:**

The client scale considered in the experiments is quite small. Cross-device federated learning typically involves a significantly larger client scale to ensure the robustness and generalizability of the results. Therefore, it would be beneficial to consider expanding the client scale in experiments to better reflect practical scenarios and to validate the scalability of the proposed method.

**Questions:**

Please refer to the weaknesses.

**Limitations:**

The experiment should include performance under varying conditions to ensure the method has no limitations in this aspect, such as different numbers of training rounds.

---

> ### Author Rebuttal · Authors · 2024-08-04
>
> Dear Reviewer LD7P:
>
> Thank you for your encouraging comments on the coordination of our methods and the overall significance of our work. We hope that our responses below will address your concerns and reinforce your positive evaluation.
>
> ### Weakness
>
> **W1: The client scale in the experiments is too small to reflect practical scenarios.**
>
> A1: For performance analysis under larger client scales, our method is extended to scenarios where each client possesses a portion of a dataset and different clients may have different datasets.
>
> To demonstrate the performance of FedSSP under varying client scales, we conducted experiments in SM cross-dataset settings with client scales ranging from 7 to 77 where each of the seven small molecule datasets was split into 1-11 segments. The results in the following Table denote that FedSSP consistently outperformed the other methods across all scenarios.
>
> *Table: **Performance under varying client scales***
>
> | Client Number | 7    | 21   | 35   | 49   | 63   | 77   |
> |:---:| :--:|:---:|:---:|:---:|:---:|:---:|
> | FedAvg       | 74.12 | 71.92 | 70.75 | 71.16 | 73.66 | 74.03 |
> | FedStar   | 78.63 | 76.37 | 74.18 | 73.20 | 75.81 | 78.28 |
> | FedSSP     | **79.62** | **76.74** | **74.86** | **74.31** | **76.25** | **79.44** |
>
> ### Limitations
>
> **L1: The experiment lacks performance evaluation under varying conditions, such as different numbers of training rounds.**
>
> A2: To demonstrate the performance of FedSSP under varying local training rounds, we conducted experiments in SM cross-dataset scenarios under different local training rounds. The results in the following Table denote that FedSSP consistently outperforms other methods in all four settings of local training rounds.
>
> *Table: **Performance under varying local training rounds ***
>
> | Local Training Round | 1  | 2 | 3 | 4 |
> |:---:| :--:|:---:|:---:|:---:|
> | FedAvg        | 74.12 | 75.56 | 74.75 | 73.81 |
> |FedStar    | 78.63 | 79.35 | 79.42 | 79.04 |
> | FedSSP       | **79.62** | **80.78** | **80.13** | **79.56** |

---

> ### Author Response · Authors · 2024-08-11
>
> Dear Reviewer LD7P,
>
> We sincerely appreciate the time and expertise you have devoted to reviewing our submission. Acknowledging the demands on your schedule, we are mindful not to intrude on your time. However, we would be grateful if you confirm that our rebuttal has addressed your concerns adequately.
>
> Thank you in advance for your consideration.
>
> Authors

---

> ### Comment · Reviewer_LD7P · 2024-08-13
>
> Dear authors,
>
> I've read all the replies and decided to keep my scoring.

---

### Decision · Program_Chairs · 2024-09-25

**Decision:**

Accept (poster)

**Comment:**

This paper focuses on structural heterogeneity in personalized Federated Graph Learning. Specifically, it effectively addresses the domain structural shift from a spectral perspective, tackling issues such as knowledge conflict and spectral bias. The framework includes Generic Spectral Knowledge Sharing (GSKS), which shares generic spectral knowledge across clients within certain components, while applying Personalized Graph Preference Adjustment (PGPA) to meet the preferences of clients for more personalized results. The experimental settings are appropriate to demonstrate the superiority of the proposed methods, although some limitations (e.g., number of clients, non-i.i.d. scenarios) were highlighted by the reviewers. The reviewers initially noted both strong points and concerns.

Strong points:

- All reviewers praise the clear motivation. Alleviating knowledge conflicts from a spectral perspective is a novel and important challenge in personalized Federated Graph Learning.

- The method is thoughtfully designed, and its performance, as well as the harmony between modules, is commendable.

- Comprehensive experiments demonstrate the effectiveness of the proposed method.

Weak points:

- Analysis of the scalability with respect to the number of clients, performance in non-i.i.d. settings, and communication cost is required.

- Additional explanation of the motivation experiments (Figure 1(c)) and justifications for sharing the filter encoder are needed.

- A more detailed discussion of related works is necessary.

The authors have adequately addressed the reviewers’ concerns. Please be sure to incorporate the rebuttal into the revised version, particularly the clarification of the motivation behind sharing the filter encoder and the detailed explanation of the motivation experiment shown in Figure 1. Moreover, the methodology section should be more refined (e.g., where is the parameter for the eigenvalue encoder (i.e., $\theta^e$) used? How is $B$ calculated?). Lastly, it would be beneficial to include an experiment that effectively demonstrates that generic knowledge has less spectral bias than other knowledge, similar to the motivation experiment in Figure 1(c). Despite some limitations, this paper has strong points that address important challenges from a novel perspective (i.e., spectral), and it holds significant potential to provide valuable insights for future work in graph federated learning.